# Nutritional Supplements and Lipedema: Scientific and Rational Use

Roberto Cannataro [1,2,3] and Erika Cione [1,2,*]

1   Galascreen Laboratories, University of Calabria, 87036 Rende (CS), Italy
2   Department of Pharmacy, Health and Nutritional Sciences, University of Calabria, 87036 Rende (CS), Italy
3   Research Division, Dynamical Business & Science Society—DBSS International SAS, Bogotá 110311, Colombia
*   Correspondence: erika.cione@unical.it

**Abstract:** Lipedema is a pathology of the adipose tissue, in evident female prevalence, diagnosed clinically and still of not well-defined etiopathogenesis. Indeed, an estrogen-related component is present, and an inflammatory state and a condition of edema are present in most cases; even pain seems to be a recurring feature, and insulin resistance is also often associated with lipedema. The therapeutic approach is finally becoming holistic. Therefore, with surgery, physiotherapy, and elastic compression therapy, the nutritional aspect of food supplementation is gaining much value. The objective of the present work is to consider the nutritional supplements that could be useful to manage this condition, underlining that, at the moment, the specific literature is practically non-existent. The most promising supplements seem to be omega 3 fish oil, polyphenols, and vitamin C, but the need for studies in this sector is urgent.

**Keywords:** lipedema; antioxidant; polyphenols; omega 3 fatty acid; inflammation

## 1. Introduction

Lipedema is a chronic pathology of the subcutaneous adipose tissue that is still not well defined by etiopathogenesis, and no diagnostic-specific biomarkers have been found. The diagnosis is clinical even though the first works date back to more than 80 years ago [1]. Lipedema is classified into five types (based on body location) and four stages (depending on severity) [2]. It presents an abnormal accumulation of adipose tissue, often in the lower limbs but also in the upper ones and on the abdomen; probably, the normal functionality of the adipose tissue is also altered. For example, Felmerer et al. [3] show how there is an alteration of the intercellular architecture (fibrosis and hypertrophy), and infiltration of macrophages is also noted, with different characteristics of both obesity and lymphedema, which can suggest an inflammatory component. Genetic predispositions are not being ruled out since the expression level of estrogen receptor [3,4] in lipedema biopsies resulted impaired. Besides that, an alteration of adipogenesis and adipocyte differentiation are present in those subjects [5]. The symptomatology is often tiredness in the lower limbs with hurt. These symptoms are present in most cases, suggesting systemic inflammation, confirmed by an alteration of the CRP serum levels. Even if CRP is not permanently elevated, it could probably evoke, from a biochemical point of view, the activation of nuclear factor kappa-light-chain-enhancer of activated B cells (NFkB), and consequently, an overproduction of Reacting Oxygen Species (ROS); the edema component is often present with the possible compression of the nerve fibers and resultant aggravation of the painful state [6]. The general condition has an important impact on the quality of life as, especially in the advanced stages (from the second onwards), even simple daily gestures are more difficult if not impossible; last but not least, the psychological component, as even in subjects who are not overweight the physical aspect is disproportionate. For those reasons the approach should be multisectoral; for example, although liposuctive surgery is effective, it does not solve the problem in a definitive way, even though it considerably improves the

patient's living conditions, but in many countries these interventions are wholly borne by the patient with very high costs, considering the need to have up to four to five surgeries. Physiotherapy and elastic-compression therapy also provide valid treatment tools. In any case, there is no definitive cure, nor effective pharmacological treatment, so the nutritional aspect and possibly food supplementation can have a vital relevance especially in the long-term management of the lipedema. Our aim is to propose nutritional supplements that can be effective, judging them on the basis of what is known about lipedema (not yet characterized at best) and, following a scientific rationale, also using scientific works on pathologies that may have aspects in common with lipedema; all this also considering our decades of experience in managing this pathology, hoping that this can be a stimulus to produce specific works. Last but not least, to try to have a rational use of nutritional supplements, given the that the overall management of lipedema is costly and often wholly left to the patient.

## 2. Nutritional Supplements

Nutritional supplements have had a considerable boost in sales in the last 20 years [7] due to a greater awareness of possible users and thanks to the push of the companies that offer them. However, unfortunately, this has meant improper uses have resulted in poor results and sometimes severe side effects, even from safe supplements [8]. Therefore, the purpose of this review (in general, our group works to provide elements that can be useful in the rational and scientific use of food supplements) is to report the current scientific literature on that topic. Currently, no specific literature directly links nutritional supplements to lipedema, so that we will propose commonly used ones with a scientific rationale, perhaps to be verified with specific studies. We want to stress the lack of specific literature on this topic. Herein, we will analyze and propose micronutrients and molecules that could benefit lipedema symptomatology.

## 3. Vitamin C

Vitamin C could have a double positive action on lipedema management. First, it is well known that ascorbic acid has an antioxidant action. Therefore, it is valuable for the inflammatory state that follows from lipedema [9]; second, another function to remember of vitamin C is the support in the synthesis of collagen [10,11]. These two biochemical actions can be helpful in the condition of lipedema that also affects connective type tissue. Recently, Bai et al. demonstrated how high doses of vitamin C could modulate pain in diabetic neuropathy [12]. In some ways, this condition could mirror lipedema's pain. Therefore, we used vitamin C in a case report, unique in lipedema, with a particularly favorable outcome of 1 g per day in doses of 500 mg of ascorbic acid [13].

## 4. Polyphenols

Like vitamin C, any substance that may have an antioxidant action can be beneficial in the management of lipedema; in this sense, thanks to their chemical structure, practically all polyphenols have an antioxidant action [14]. Those that have a modulatory action on NFkB would be preferred, as this cellular mediator, if activated, has a powerful action on the synthesis of inflammatory mediators and free radicals. It is known that in osteoarthritis the NFkB pathway is upregulated [15–17]. Since this is also a connective localization, a similar involvement could also be assumed in lipedema. In any case there is a strong inflammatory component, often the CRP is also altered [5]. Again, this is our hypothesis that should be verified; certainly an inflammatory component and consequent production of ROS seems very consistent with lipedema. For example, we have shown [18] how in particular a group of polyphenols typical of the Mediterranean diet are capable of downregulating the NFkB pathway, also underlining its action on miRNAs, small nucleotide sequences capable of regulating protein synthesis. Among these, oleuropein is one of the compounds that should be proposed. In fact, in our study we strongly recommended extra virgin olive oil (EVO) as a seasoning fat [13]. When it is not possible, for example far from

the Mediterranean basin where EVO is difficult to find if not at a high cost, we could suggest the integration of oleuropein or its derivatives such as hydroxytyrosol. It has been seen that it has a strong antioxidant action, not only thanks to the action on the NFkB pathway [14,17]. Curcumin has also proved effective in this sense, both as an antioxidant in its own right and as a regulator of NFkB, plus it boasts various trials in the management of chronic diseases, as is lipedema [14]. Another mechanism activated by curcumin is that of nuclear factor erythroid 2-related factor 2 (Nrf2) [18], a mediator which instead has a positive effect on the regulation and management of ROS. This too, obviously, would be very positive in the managing of lipedema. In their work, Coletro et al. [19] show how diets rich in polyphenols and possibly a supplementation have an important efficacy in mitigating the effects, and in particular the painful component, in rheumatic diseases such as osteoarthritis (which as mentioned has some overlapping facets to lipedema), fibromyalgia (often also present in patients with lipedema), and rheumatoid arthritis, improving pain in almost 65% of cases and inflammation in 58%. Another interesting aspect, at least in part can be ascribed to the anti-inflammatory action of polyphenols and the positive action on polycystic ovary syndrome, which often occurs in conjunction with lipedema. Mihanfar et al. [20] show how a dietary intake of polyphenols can be help in mitigating the effects of this syndrome**.** Maity et al. [21] underline the importance of the contribution of polyphenols in the management of rheumatoid arthritis, also proposing innovative delivery methods. In fact, for more than one case the proven in vitro activity is not reflected in vivo due to poor bioavailability, therefore our advice is to strongly suggest a diet rich in polyphenols, recommending a style close to the Mediterranean diet. In any case, we recommend to ensure a daily intake of 100–150 mg of polyphenols, coming from multiple sources, as each molecule has a union different genomics or epigenetics and often in synergy with the others. Where this cannot be achieved, one can think of integrating the contribution with extracts, possibly from different plants.

## 5. Omega 3 Fatty Acids

The anti-inflammatory activity of omega 3 fatty acids is well supported by the scientific literature [22,23]. First of all, it must be emphasized that products or foods containing linolenic acid are often mistakenly considered effective in this sense. From a chemical point of view the latter is part of the family of omega 3 fatty acids, as its first double bond C = C is located at the third carbon, but it has no anti-inflammatory action and its conversion into ecosapentaenoic acid (DHA) and docosaheptaenoic (EPA) is poor. Therefore, to have the desired effect it is necessary to take EPA and DHA directly. The inflammatory component in lipedema is very relevant, therefore the action of DHA and EPA could be very useful. Other than a case report we have published [13], there are no manuscripts operated directly on lipedema. In the review by Kuda et al. [24] it is emphasized that an integration of DHA and EPA can promote the health of adipocytes, even in obese subjects, where there is subclinical inflammation, through a mechanism that involves a lower activation of macrophages and consequent lower secretion of pro-inflammatory cytokines. In addition, the synthesis of mediators that modulate inflammation such as resolvin and maresin, and in general the family of protectins, synthesized starting from DHA and EPA, is also well established. The mechanism of action is not yet clear, but probably involves the Transient Receptor Potential (TRP) channel. In this sense it is interesting to note how probably different resolvins modulate the action of different channels (e.g., TRPA1, V1, V4) [25,26]. This could be a very important feature as at least 80% of subjects affected by lipedema manifest a painful component. In fact, we agree with the review by Sun et al. [27], who would like to see at least DHA as an essential nutrient given the numerous health features. Specifically, we would like to recommend a daily intake of at least 1 g of DHA and EPA, which could be also doubled at an early stage, to moderate inflammation and painful component.

## 6. Oedema Modulating Agents

*6.1. Serratiopeptidase*

Serratiopeptidase or serrapeptase is a proteolytic enzyme produced by the bacterium Serratia Marcescens, stable in a range of pH between 3 and 10, with maximum activity at 9, therefore sensitive to gastric acidity. For this reason, the dosages used are pretty high [28,29]. Although it is widely used, especially in traditional Indian medicine, a mechanism of action is not recognized; the anti-inflammatory action, although reported, does not seem to have a consolidated scientific rationale, while the most plausible effect, potentially useful in the management of lipedema, could be fibrinolytic and in general proteolytic [29]. In fact, an edematous component is often present in subjects suffering from lipedema, so it could prove helpful. Although there have been two reports with adverse effects [30,31], even major ones, serratiopeptidase appears to be relatively safe in dosages up to 2 g per day. Bioavailability, also due to possible sensitivity to gastric pH, is not very high. In fact, the best results are shown when it is used locally as a result of surgical interventions in the mouth [28,32], where the gastric passage is bypassed. In this sense, liposomal and possibly sublingual formulations are being developed [33]. Our opinion is that this molecule, although not dangerous, needs more evidence to be used in general, and in lipedema in particular.

*6.2. Bromelain*

Bromelain is a proteolytic enzyme characteristic of pineapple. The first studies on its use more than 50 years ago showed a fibrinolytic and degreasing action on clots. In this sense, it could be considered in the management of lipedema. However, there are no specific studies to that end. In recent years, an immunomodulating action is also being considered in addition to antimicrobial activity. This action could also be useful as in lipedema, as due to inflammation, there is a marked activation of the immune system, at least locally [34]. Bromelain is a widely used supplement, therefore, relatively safe. On the other hand, the effectiveness is to be tested because, despite various hypotheses, the mechanism of action is unclear.

## 7. Miscellaneous

In this paragraph, we report other supplements commonly used to try to support the management of lipedema. However, often there is no solid scientific rationale and/or their use comes from anecdotal experiences.

*7.1. Vitamin D*

The pleiotropic action of the vitamin is now well established, as well as the negative correlation with the accumulation of fat, as often, in the 25 (OH) D form it is trapped in the inflamed adipose tissue, reducing the quantity of the biologically active one [35]. The levels of vitamin D influence even the normal state of health of the adipose tissue, therefore it would be advisable, after evaluating the plasma levels, to supplement with vitamin D; for example, in our case report integration was found to be necessary.

*7.2. Vitamin B12*

Since this vitamin has been isolated, the beneficial effect on the nervous system, particularly the peripheral one, has been highlighted. In lipedema, a painful neuropathic component is becoming increasingly evident. Therefore vitamin B12 could be beneficial. In the review by Buesing et al. it is highlighted that it is also effective in the treatment of pain, in particular, neuropathic pain. It must also be emphasized that the analysis of plasma cobalamin levels may not be the mirror of reality. For this vitamin, the value of plasma levels would be desirable to stay in the central or upper part of the normal range. It is worth remembering the normal ranges are 160 to 950 picograms per milliliter (pg/mL), and 118 to 701 picomoles per liter (pmol/L) [36,37], although it would be better to also

evaluate the levels of holotranscobalamin (holoTC); based on plasma values we suggest supplementation of 500–100 mcg.

### 7.3. Magnesium

This supplement is often used by those suffering from lipedema. It cannot be considered as effective, as it does not in itself have a direct action on the complications of lipedema. The same applies to this mineral as vitamin D, i.e., it is advisable to restore its ranges of normality, even if, unlike vitamin D, the analysis of the blood content is not a good evaluation index.

### 7.4. Selenium

In a retrospective cross-sectional study, Pfister et al. [38] analyzed 198 subjects with lipedema and 168 subjects with Lipo-Lymphedema (they screened other subjects with lymphedema), noticing a selenium deficiency. However, the deficiency of this microelement is also frequent in the healthy population and significantly related to the area, so we do not think it can be a supplement to be considered in any case for lipedema. It should be emphasized that selenium is very important for the correct functioning of the immune system and for the management of free radicals; therefore, as indicated for vitamin D, we recommend perhaps to evaluate its plasma values and consequently think of integration if necessary.

### 7.5. Butcher's Broom

Ruscus Genus it is a typical Mediterranean vegetable, it has been used for centuries as a traditional medicine for the treatment of wounds and ulcers; in fact, at the moment, it is used to manage venous insufficiency and other pathologies affecting the circulatory system [39,40]. It is recommended as a support in the treatment of lipedema, but the component affecting the venous system is often not marked. It is proved to be safe, practically no adverse effects are reported, but its use in lipedema seems to be inappropriate, even if in case reports combined with selenium and a physiotherapy treatment it gave good results [41].

### 8. Conclusions

Lipedema is a pathology that still needs evaluation and studies to be characterized. Certainly, a multifactorial approach must be considered. One of these factors is undoubtedly nutrition, to be completed or enriched with nutritional supplements. Obviously this does not mean curing the pathology, but mitigating some aspects, in particular the painful manifestations probably due to an inflammatory state. We report in Table 1 those we have considered, underlining that to date there are no specific studies concerning lipedema. Therefore, according to our experience "in the field" and according to a biochemical and physiological rationale, but also trying to consider the works carried out on pathologies that may have points in common with lipedema, we have given guidelines on what could be nutritional supplementation in support of lipedema. Studies should undoubtedly be carried out to verify its effectiveness. On the other hand, as in various situations, supplements are used, at least in a redundant way, so we recommend the use of omega 3, vitamin C in any case, polyphenols, vitamin D, vitamin B12 and magnesium after checking the plasma values and/or the contribution in the diet. We hope that the present manuscript could provide insight to produce specific literature. It is not easy to organize interventional studies, as to prove the effectiveness of a supplement it would be necessary to use it exclusively without operating other treatments. Therefore, the recruitment of subjects for an interventional and non-observational study is not easy. On the other hand, the incidence of lipedema seems to be increasing (probably due to the increase in diagnoses), so it is a problem that affects more and more people. Finally, a possible recommendation to specialized personnel who manage lipedema is to use a few supplements with a solid scientific rationale behind them and perhaps to make public any good results coming from daily practice.

**Table 1.** Here we report the supplements and their use based on the current literature Y: suggested, E: evaluate its use, N: not suggested.

| Supplement | Biochemical Action | Dosage | Suggestion |
|---|---|---|---|
| DHA and EPA | Anti-inflammatory Pain relieving | 1–2 g per day | Y |
| Vitamin C | Antioxidant Support on collagen synthesis | 500–1000 mg per day | Y |
| Polyphenols | Anti-inflammatory Antioxidant | 100–200 mg per day | E |
| Vitamin D | Adipose tissue health Immunomodulation | 2000 IU per day | E |
| Vitamin B12 | Pain relieving Neuropathy treatment | 500–1000 mcg | E |
| Magnesium | General Health Muscle trophism Pain management | 300–400 mg per day | E |
| Selenium | Immune system ROS scavenging | 45–60 mcg per day | E |
| Serratiopeptidase | Oedema management Anti-inflammatory | 200–400 mg per day | N |
| Bromelain | Oedema management | 300–600 mg per day | N |
| Butcher's Broom | Vessel trophism Lymphatic circulation | 150–300 mg per day | N |

**Author Contributions:** E.C. and R.C. equally contributed to the whole process of the manuscript. All authors have read and agreed to the published version of the manuscript.

**Funding:** This research received no external funding.

**Institutional Review Board Statement:** Not applicable.

**Informed Consent Statement:** Not applicable.

**Acknowledgments:** The authors want to thank LIO lipedema, Italian Lipedema Association for the support.

**Conflicts of Interest:** The authors declare no conflict of interest.

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
