# Peer review of "Nutritional Supplements and Lipedema: Scientific and Rational Use"

_nutraceuticals, doi:10.3390/nutraceuticals2040020_

Round 1

Reviewer 1 Report (Previous Reviewer 2)

          The title of this paper need to be revised. It does not reflect the objectives of this study.

          The main aim of this review has not been presented in the introduction portion. It should be present at the last of introduction portion. It must justify the work.

          Authors have used sometimes lipoedema and sometimes lipedema, CORRECT IT?

          It lacks proper rhythm.

          The title of this paper need to be revised. It does not reflect the objectives of this study.

Author Response

In the present manuscript, the authors summarize the potential beneficial effects of nutritional supplements on lipoedema. Although, there is no specific literature, authors propose several nutritional supplements that are usually used and may have a scientific rationale. Among the mentioned supplements, omega3 fish oil, polyphenols, and vitamin C are suggested to be used. It seems like that nutritional supplements that have anti-inflammatory effects may be potentially used in patients with lipoedema. Several concerns about the manuscript that are required to be addressed. Authors should introduce the pathology of lipoedema in more detail. What is the urgent research direction of lipoedema? What are the obstacle to treatment in clinical? Authors recommend use of nutritional supplements in lipoedema. However, lack of specific literature supports the point. In the manuscript, supplements are classified as antioxidants, anti-inflammatory supplements, and oedema modulating agents. Please explain the significance of anti-oxidative potentials and NFkB upregulation in lipoedema. Several format errors are found in the manuscript, please check and correct. 

We thank the reviewer for the usefull suggestions

We better characterized the pathology of lipedema

We reorganized the presentation of supplements, and we better-checked errors

Reviewer 2 Report (Previous Reviewer 1)

This manuscript speculates that appropriate nutritional supplements can improve lipedema. They recommend that omega 3 and vitamin C can be used in any case and that polyphenols, vitamin D, vitamin B12, and magnesium can be used appropriately after checking plasma values and dietary contributions. As the authors state, these supplements to support lipedema management often have no solid scientific basis but rather anecdotal experience, so use should be done with skepticism, and more research is needed to confirm. Based on the current content, I think the author can add more detailed information on lipedema, as well as ways to identify lipedema clinically. In addition, please explain how you know that lipedema can be improved by giving nutritional supplements? In addition, the subtitle numbers of Sections VI and VII need to be revised.

Author Response

We thank the reviewer fro the useful suggestions.

Abstract, title and references

  • The aim of this study is clearly reported as to study the use of nutritional supplements that could be useful to manage the lipoedema.
  • The study showed that at the moment, the specific literature is practically non-existent. The most promising supplements seem to be omega3 fish oil, polyphenols, and vitamin C.
  • The title of this paper need to be revised. It does not reflect the objectives of this study.

References

All the references are relevant, recent and referenced correctly.

All the references have been found in the reference list below.

Introduction/background

  • Authors have clearly mentioned the background of the review in the introduction portion.
  • The main aim of this review has not been presented in the introduction portion. It should be present at the last of introduction portion. It must justify the work.

We changed the title, and we expanded the introduction

Discussion and Conclusions

  • The authors have collected very less literature to justify the objective of this study
  • It lack proper rhythm. Polyphenols comes under two sub titles.

Recommendation: Why not to discuss under single title.

  • The previous findings have been discussed in multiple angles using different sources of references and have been placed in the context without being interpreted.
  • Conclusion justifies the main objectives of this review.
  • References have been used to support this study.

We expanded citations, we condensed polyphenols into one section.

Specific comments on weaknesses of the article and what could be done to improve it

Major points in the article which needs clarification, refinement, reanalysis, rewrites and/or additional information and suggestions for what could be done to improve the article. 

  1. Authors have used sometimes lipoedema and sometimes lipedema, CORRECT IT?
  2. It lacks proper rhythm. Polyphenols comes under two sub titles.
  3. The authors have collected very less literature to justify the objective of this study.
  4. The main aim of this review has not been presented in the introduction portion. It should be present at the last of introduction portion. It must justify the work.
  5. The title of this paper needs to be revised. It does not reflect the objectives of this study.

1 we used only lipedema

2 we condensed polyphenol section

3 we expanded references

4 we expanded the introduction

5 we changed the title, but if you have any suggestions....

Reviewer 3 Report (New Reviewer)

The aim of the paper was to review literature on nutritional supplements in lipoedema.  The review is clear, concise and relevant to the field, yet I feel like it is not as comprehensive as it could be. E.g. authors do not refer to Lypmhedema and Lipedema Nutrition Guide (by Ehrlich, Iker etc) that discusses a vast number of supplements. I would be curious to know what is authors stance towards the examples presented in the book or why they omitted it. They also do not mention Nourollahi, S., Mondry, T. E., & Herbst, K. L. (2013). Bucher’s Broom and selenium improve lipedema: A retrospective case study. Altern Integ Med2(119), 2. and I would be curious to know their stance on that study as well.    Also Selenium that is not even mentioned in the current review: Pfister, C., Dawczynski, H., & Schingale, F. J. (2020). Selenium Deficiency in Lymphedema and Lipedema—A Retrospective Cross-Sectional Study from a Specialized Clinic. Nutrients12(5), 1211.     Yet I'd recommend to provide specific title for Table 1, clearly showing that these are authors recommendations or suggestions based on the existing literature.   In the conclusions authors are writing that they are  "hoping that the present manuscript could be insight to produce specific literature". I assume they are meaning that there is a need for further research. I would be interested to see if they have any specific research ideas they would like to present in the conclusions part.    I also recommend to do an English language proofreading. 

Author Response

The aim of the paper was to review literature on nutritional supplements in lipoedema.  The review is clear, concise and relevant to the field, yet I feel like it is not as comprehensive as it could be. E.g. authors do not refer to Lypmhedema and Lipedema Nutrition Guide (by Ehrlich, Iker etc) that discusses a vast number of supplements. I would be curious to know what is authors stance towards the examples presented in the book or why they omitted it. They also do not mention Nourollahi, S., Mondry, T. E., & Herbst, K. L. (2013). Bucher’s Broom and selenium improve lipedema: A retrospective case study. Altern Integ Med2(119), 2. and I would be curious to know their stance on that study as well.    Also Selenium that is not even mentioned in the current review: Pfister, C., Dawczynski, H., & Schingale, F. J. (2020). Selenium Deficiency in Lymphedema and Lipedema—A Retrospective Cross-Sectional Study from a Specialized Clinic. Nutrients12(5), 1211.     Yet I'd recommend to provide specific title for Table 1, clearly showing that these are authors recommendations or suggestions based on the existing literature.   In the conclusions authors are writing that they are  "hoping that the present manuscript could be insight to produce specific literature". I assume they are meaning that there is a need for further research. I would be interested to see if they have any specific research ideas they would like to present in the conclusions part.    I also recommend to do an English language proofreading. 

We thank the reviewer for positive comment.

We read carefully the book cited, but most of the supplements mentioned do not have solid scientific literature, so we decided to include only those that possess a solid background.

We did not mention the case report because the use of supplements is not exclusive, i.e., the pleasing result is not attributable to the supplements used, anyway, we added to the manuscript

We missed the other paper suggested we included in the review, even, as noted by the authors, selenium deficiency is common, so it could be considered but not used in any case on lipedema.

We suggested future studies, even though it is difficult to perform them.

Round 2

Reviewer 2 Report (Previous Reviewer 1)

This revised manuscript was improved and can be accepted now. 

This manuscript is a resubmission of an earlier submission. The following is a list of the peer review reports and author responses from that submission.

Round 1

Reviewer 1 Report

In the present manuscript, authors summarize the potential beneficial effects of nutritional supplements on lipoedema. Although, there is no specific literature, authors propose several nutritional supplements that are usually used and may have a scientific rationale. Among the mentioned supplements, omega3 fish oil, polyphenols, and vitamin C are suggested to be used. It seems like that nutritional supplements that have anti-inflammatory effects may be potentially used in patients with lipoedema. Several concerns about the manuscript that are required to be addressed. Authors should introduce the pathology of lipoedema in more detail. What is the urgent research direction of lipoedema? What are the obstacle to treatment in clinical? Authors recommend use of nutritional supplements in lipoedema. However, lack of specific literature supports the point. In the manuscript, supplements are classified as antioxidants, anti-inflammatory supplements, and oedema modulating agents. Please explain the significance of anti-oxidative potentials and NFkB upregulation in lipoedema. Several format errors are found in the manuscript, please check and correct. 

Reviewer 2 Report

Abstract, title and references

          The aim of this study is clearly reported as to study the use of nutritional supplements that could be useful to manage the lipoedema.

          The study showed that at the moment, the specific literature is practically non-existent. The most promising supplements seem to be omega3 fish oil, polyphenols, and vitamin C.

          The title of this paper need to be revised. It does not reflect the objectives of this study.

References

All the references are relevant, recent and referenced correctly.

All the references have been found in the reference list below.

Introduction/background

          Authors have clearly mentioned the background of the review in the introduction portion.

          The main aim of this review has not been presented in the introduction portion. It should be present at the last of introduction portion. It must justify the work.

Discussion and Conclusions

          The authors have collected very less literature to justify the objective of this study

          It lack proper rhythm. Polyphenols comes under two sub titles.

Recommendation: Why not to discuss under single title.

          The previous findings have been discussed in multiple angles using different sources of references and have been placed in the context without being interpreted.

          Conclusion justifies the main objectives of this review.

          References have been used to support this study.

Specific comments on weaknesses of the article and what could be done to improve it

Major points in the article which needs clarification, refinement, reanalysis, rewrites and/or additional information and suggestions for what could be done to improve the article. 

1.              Authors have used sometimes lipoedema and sometimes lipedema, CORRECT IT?

2.              It lacks proper rhythm. Polyphenols comes under two sub titles.

3.              The authors have collected very less literature to justify the objective of this study.

4.              The main aim of this review has not been presented in the introduction portion. It should be present at the last of introduction portion. It must justify the work.

5.              The title of this paper need to be revised. It does not reflect the objectives of this study.